# Patient and Public Involvement in Malnutrition Disorders Health Research: A Methodological Systematic Review Protocol

**DOI:** 10.3390/healthcare13070839

**Published:** 2025-04-07

**Authors:** Arturo Garcia-Garcia, Cristina Carretero-Randez, Rafaela Camacho-Bejarano, María Teresa Roldán-Chicano, Pedro Raúl Castellano-Santana, Lucía Rocío Camacho-Montaño, Jesica Montero-Marco, Marta Charlo-Bernardos, María Isabel Orts-Cortes

**Affiliations:** 1Research Group Quality of Life and Health, Department of Nursing, Faculty of Health Sciences, European University of Valencia, 03016 Alicante, Spain; 2Hospital Emergency Department, Hospital University Vinalopó, 03202 Elche, Spain; 3Nursing Research Group: Infection, Inflammation and Chronicity (IIS La Fe), Hospital University and Polytechnic La Fe, 46026 Valencia, Spain; carreterorandez.cristina@gmail.com; 4Community Health and History of Science Team, Nursing Department, University of Alicante, 03690 Alicante, Spain; 5Department of Nursing, Nursing Faculty, University of Huelva, Campus El Carmen, 21071 Huelva, Spain; rafaela.camacho@denf.uhu.es; 6Nursing and Healthcare Research Unit (Investén-Isciii), Institute of Health Carlos III, 28029 Madrid, Spain; lcamacho@isciii.es (L.R.C.-M.); isabel.orts@ua.es (M.I.O.-C.); 7Research Network on Chronicity, Primary Care and Health Promotion (RICAPPS), 37005 Salamanca, Spain; 8Center for Research in Contemporary Thought and Innovation for Social Development (COIDESO), University of Huelva, 21007 Huelva, Spain; 9Research Unit Care, Area 2 of the Murcian Health Service, 30202 Cartagena, Spain; mteresa.roldan@um.es; 10Department of Nursing Faculty, University of Murcia, Cartagena Campus, 30001 Murcia, Spain; 11Complejo Hospitalario Universitario Materno-Insular de Gran Canaria, 35016 Las Palmas, Spain; pedroraul.castellano@ulpgc.es; 12Department of Nursing, Nursing Faculty, Universidad de Las Palmas de Gran Canaria, 35201 Las Palmas, Spain; 13Research Unit, Hospital Clínico Universitario Lozano Blesa, Av. San Juan Bosco, 15, 50009 Zaragoza, Spain; jmontero@salud.aragon.es; 14GIIS081-Care Research Group, Instituto de Investigación Sanitaria Aragón (IIS Aragón), Av. San Juan Bosco, 13, 50009 Zaragoza, Spain; mcharlo@salud.aragon.es; 15Continued Training Unit, Hospital Clínico Universitario Lozano Blesa, Av. San Juan Bosco, 15, 50009 Zaragoza, Spain; 16Department of Nursing, Alicante Institute for Health and Biomedical Research (ISABIAL), University of Alicante (BALMIS), 03690 Alicante, Spain; 17CIBER of Frailty and Healthy Ageing (CIBERFES), Institute of Health Carlos III, 28029 Madrid, Spain

**Keywords:** patient and public involvement, malnutrition, community-based participatory research, patient participation, health services research

## Abstract

**Background/Objectives**: Older adults are particularly susceptible to undernutrition and conditions that can aggravate it, such as frailty and conditions associated with swallowing difficulties or dysphagia. To address these challenges, it is important to consider the perspectives of older adults and their caregivers, especially those with conditions such as frailty or cognitive impairment, as they can provide valuable insights on supporting nutrition in these vulnerable populations. This participatory approach requires structures formed by scientific research committees working together with other stakeholders, involving various actors at all stages of the research process. The aim of this study is to analyze the methodology for involving patients aged 65 and older with malnutrition or at risk of malnutrition as co-investigators in research. **Methods**: This protocol has been developed in accordance with the Preferred Reporting Items for Systematic Reviews and Meta-Analyses (PRISMA-P) checklist. A literature search will be carried out in the following electronic databases: PubMed/MEDLINE, EMBASE and CINAHL. Through the COVIDENCE program, the research team will independently review the different screening phases of the identified studies for possible inclusion or exclusion. **Expected Results**: This systematic review will provide up-to-date evidence on the use of non-scientific actors at different stages of research. The main limitation stems from the use of non-scientific agents in a topic as specific as adults with or at risk of undernutrition, which may make it difficult to extrapolate the results to other settings. The registration number in PROSPERO is CRD42024444374.

## 1. Introduction

Malnutrition encompasses both overnutrition and undernutrition. Recent demographic studies indicate that up to 25% of older adults may be undernourished or at risk of undernutrition [1,2]. The rapid growth of the aging population, combined with age-related changes and specific nutritional requirements, further contributes to malnutrition risk [3].

Conversely, malnutrition can contribute to premature mortality through severe outcomes such as frailty, delirium, reduced immune function, muscle wasting, and cognitive impairment. These factors hinder recovery during illness and diminish patients’ quality of life. Therefore, it is essential to properly assess eating and swallowing disorders using validated clinical and nutritional tools that enable early detection and the implementation of individualized interventions (an-thropometric measurements, biochemical parameters, MUST, GLIM, EAT-10, among others) [1,4].

Current estimates suggest that about a quarter of older adults are undernourished or at risk of undernutrition, and this number is expected to increase with the rapid growth of the aging population. The increase in the risk of malnutrition may be due to physiological changes associated with aging and specific nutritional needs, among other factors. Lack of resources, inaccessible and insufficient information from professionals on malnutrition are some factors that may explain the high rates of disease-related malnutrition [4]. Tailored counseling, person-centered care, active implementation strategies, are needed to address this problem [1]. In this regard, it is crucial to incorporate the perspectives of patients and caregivers across a range of conditions, including but not limited to frailty or cognitive impairment, as they can offer diverse insights into supporting older adults’ nutritional needs. Understanding their input would help inform the development of appropriate and acceptable services to reduce the risk of malnutrition [2].

A participatory approach requires structures formed by scientific research committees working alongside other stakeholders, involving various actors at all stages of the research process [5]. Engaging non-scientific actors from the community and co-developing research is essential in order to identify the strengths, priorities, and needs of the community, and to translate the results into policies, practices, or actions for change [5,6].

In this protocol, ‘the community’ refers to older adults themselves—whether they are living independently or in care facilities—as well as informal caregivers, family members, patient associations, and other stakeholders directly affected by malnutrition who may not hold formal scientific roles. These community actors can still play a critical part in shaping the study design, data interpretation, and dissemination of results. It is about creating a relationship where there is bilateral communication, research is generated for future actions based on a better-defined context, there is effort from both researchers and participating actors, and co-ownership of the research and knowledge [7].

It has been demonstrated that employing this more participatory approach fosters the incorporation of diverse perspectives from stakeholders throughout the various stages of the research process. Without the inclusion of such stakeholders, the findings risk overlooking cultural contexts and nuances, leading to less robust outcomes and potentially compromising the external validity of the research. Furthermore, involving the community under study strengthens and enhances the achievement, applicability, and quality of the results. While this approach is valuable, it can also be more resource-intensive in terms of time and effort, particularly in selecting and incorporating the participants who will engage in the research [8].

Implementing this participatory approach in research signifies that individuals and communities not only provide information or act as advisors but also engage meaningfully with researchers in the design and stages of the study [9].

In 2009 and 2010, two bibliographic reviews were conducted by Staley (INVOLVE Study) and Brett et al. (PIRICOM Study) [10,11] on the impact of public involvement in research (PPI). Both reviews identified limitations related to the scarcity of primary studies and variability in the quality of published literature in this area. Additionally, they highlighted the wide variation in methods used to evaluate the impact of public participation in research, as much of the evidence relied on retrospective opinions of researchers and, to a lesser extent, the involved public [12].

Following these studies, support for public participation in research was initiated in Spain through the National Institute of Health Research. The objective was to involve the public in research projects, accompanied by efforts to develop an evidence base for public involvement in health and social care research. In 2014, Evans et al. identified contextual factors (e.g., principal investigator leadership, culture, field of research) and key mechanisms (e.g., leadership roles, resource allocation, facilitation) that could facilitate successful public involvement in research. By 2017, international guidelines, such as GRIPP2 (Guidance for Reporting Involvement of Patients and the Public), were developed to improve the quality of evidence on patient and public involvement (PPI) in health and social care research. These guidelines provide checklists to enhance the reporting quality of public and patient involvement in research. However, this methodology has divided attention across scientific projects, depending on the context [12,13].

There remains a significant lack of research describing how best to include non-scientific actors, a term we use here to encompass diverse stakeholders such as family caregivers, patient associations, or community representatives. In some cases, these partners serve as co-researchers, offering consistent input throughout the project; at other times, they become co-investigators, indicating a deeper level of shared decision-making with the academic team. By clarifying these roles, we can develop and disseminate strategies that enhance researchers’ preparedness to collaborate effectively. This would enable the development of research designs that include clear descriptions and incorporation of non-scientific actors [14,15,16].

Older adults are often underrepresented in public participation in research (PPI) due to the additional barriers involved in including individuals with progressive diseases, cognitive impairments, and limitations in mobility or speech. Moreover, age-related conditions, along with the stigma associated with functional and cognitive difficulties, may lead to the perception that they are unable to contribute valuable input in research processes, inadvertently excluding them. It is precisely this underrepresentation that makes it even more urgent to involve the older population in research, as their voices are often the least heard, which impacts the equity and relevance of the results obtained. The scientific literature has progressively begun to include studies on the participation of older adults, with the development of syntheses and guidelines on methodologies for including this population. However, in reality, most studies involving older adults and malnutrition provide limited details on how non-scientific actors are recruited or integrated into the research process and no established guidelines have been found for the co-design of research involving older adults, particularly in the field of nutrition [17,18,19].

There is a growing involvement of older adults in developing research studies, which may be a promising way to address the limitations between evidence-based knowledge and current clinical practice. This approach has a positive impact not only on the quality of research but also on the confidence, learning, and activism of older co-researchers. Patient and public involvement (PPI) strategies for building relationships include scheduling regular team meetings and reflection sessions, maintaining a flexible and interactive process. However, in the context of older adults at risk of malnutrition, there are no clear guidelines on the involvement of patients as co-researchers and the application of PPI in research development [17].

By publishing this protocol, we aim to offer a clear methodological roadmap for researchers interested in systematically involving older adults at risk of malnutrition as co-investigators. While the final outcomes will be presented in subsequent publication, this protocol provides immediate value by

Ensuring methodological rigor and transparency, thus reducing the risk of selective reporting;Allowing other researchers to replicate or adapt our approach to different populations and contexts;Establishing a foundational framework for patient and public involvement (PPI) in nutritional and/or swallowing disorders research.

In doing so, we hope to enhance the quality and relevance of future research on malnutrition, bridging the gap between the academic sphere and the real-world perspectives of older adults and their caregivers.

## 2. Materials and Methods

The research question is framed using the PICO model:

P (Population): patients ≥ 65 of age with impaired nutritional status or deglutition disorders;

I (Intervention): methodologies for including non-scientific actors (PPI) in research;

C (Comparison): not applicable in the traditional sense, as we are not evaluating a control group but rather examining different PPI methods;

O (Outcome): assessment of these methods (using GRIPP2) and the degree of participant involvement

It is important to note that these questions are aimed at providing evidence to meet the objective of our study. As the review develops and evidence is gathered, these questions may be modified or developed to achieve the objective of the study.

### 2.1. Experimental Design

This protocol follows a systematic review design focusing on methodologies for patient and public involvement (PPI). We will first establish the specific objectives and the research question. Next, we will develop the search strategy for the selected databases (PubMed, EMBASE, CINAHL), choose studies based on our inclusion and exclusion criteria, conduct data extraction and quality assessment, and finally synthesize the findings. This protocol follows the Preferred Reporting Items for Systematic Reviews and Meta-Analyses for Protocols (PRISMA-P) guidelines, as illustrated in Figure 1. This figure depicts each stage of the review: (1) defining the research question and objectives, (2) developing the search strategy, (3) screening and selecting studies, (4) data extraction and quality assessment, and (5) synthesizing and concluding. Figure 1 thus provides a clear roadmap for our systematic review process. The complete PRISMA-P checklist is provided as a Appendix A for transparency [20]. The registration number in PROSPERO is CRD42024444374.

### 2.2. Detailed Procedure

Literature search: We will look through the chosen databases without language restrictions but will apply publication date limits (see Section 2.3.5. Search strategy).

Study selection: Pairs of reviewers will screen titles and abstracts using COVIDENCE.

Data extraction and quality assessment: We will apply recognized tools such as those from JBI and the Newcastle–Ottawa scale, among others.

Synthesis: We will group the included studies (quantitative, qualitative, mixed methods) and employ both a narrative approach and thematic analysis.

Results dissemination: We aim to publish in indexed journals and present at conferences to share outcomes widely.

### 2.3. Eligibility Criteria

#### 2.3.1. Types of Studies

This review will include primary research that meets quality criteria and describes the involvement of non-scientific actors in decision-making, such as co-investigators, alongside researchers during the development of the research. Quantitative studies with analytical and experimental observational designs will be included (randomized controlled, quasi-experimental, and non-randomized controlled trials, as well as cohort and case–control studies). Qualitative studies or mixed methods in which data can be extracted according to the aim of the review could be extracted. Non-original studies and other secondary studies will be excluded (comments, opinions, letters, editorials, etc.). Systematic reviews whenever they involve non-scientists in the study will be considered as additional information but will not be included.

#### 2.3.2. Population

We will include studies involving individuals aged ≥ 65 years with malnutrition or at risk of malnutrition (covering undernutrition, overweight, or other related nutritional imbalances) and/or deglutition disorders (including dysphagia).

Studies specifically involving a pediatric population or conducted in any other setting will be excluded.

Types of intervention/phenomena of interest.

The phenomenon of interest in this review will be the methodology used to include and involve non-scientific actors in various stages of a research study. We will include studies that incorporate non-scientific actors or describe the strategies used to enable their involvement in research.

#### 2.3.3. Context

Clinical settings may be varied and could include ambulatory care, outpatient follow-up clinical specialties in hospitals, community settings, or primary care in any geographic location.

#### 2.3.4. Results

The main outcome will be the involvement of non-scientific actors in research.

This outcome will be assessed based on the description of the use and degree of completion of the GRIPP2 tool (short form). Including, as key outcomes, the role played by non-scientific stakeholders, and level of involvement in the development of the research.

#### 2.3.5. Search Strategy

A systematic and exhaustive literature search will be carried out by a librarian with expertise in electronic searches, using a combination of free and controlled terms.

The following electronic databases will be searched to identify quantitative, qualitative and mixed studies: PubMed/MEDLINE, EMBASE, and CINAHL.

The main search will be conducted in each database restricting the publication dates to the last ten years (2014–2024) to ensure that we capture the most current and relevant PPI methodologies. We will include articles published up to August 2024. Studies will be considered without filtering by type of country, or language.

Thesaurus database descriptors and natural language keywords pertaining to the focus of the study will be selected by a research team who will compile and refine the search terms before conducting a full search to capture potentially relevant publications.

Search strategy in PICO format [21]:-P: Malnutrition; risk of malnutrition; deglutition disorders.-I: Patient and public involvement, Community-Based Participatory Research-C: Non-scientific, Patient Participation, Community Participation-S: Systematic reviews and meta-analyses; RCTs and CCTs; observational studies; qualitative studies.

All search strategies for all databases are included as Appendix B (Table A1).

NOT will be used to limit terms related to animals, editorials, letters, comments, conference proceedings, retractions, and books.

A hand search and citation search of the included primary studies and key journals known to the reviewers and those identified during the search will be conducted.

### 2.4. Study Selection Process

Through the COVIDENCE program (where all documents will be stored), the research team, in pairs, will independently review the titles and abstracts of studies identified for possible inclusion. Subsequently, the full text of all potential studies agreed for inclusion will be obtained and independently reviewed. Disagreements will be resolved through a third independent reviewer. Reasons for exclusion will be reported on the basis of the selection criteria described.

### 2.5. Data Extraction (Selection and Coding)

To extract data and assess the quality of the studies the research team members will work in pairs simultaneously according to their expertise in qualitative or quantitative methodology. A third member will be involved in case of discrepancy.

To extract data and evaluate the quality of the studies, members of the research team will work in pairs simultaneously according to their experience in quantitative methodology. The data extraction tool proposed by the Joanna Briggs Institute for Evidence-Based Practice (JBI 2014) is a standardized tool for extracting data from quantitative studies. Specific details about the interventions/exposures, study methods, and outcomes relevant to the review question and specific objectives will be included. We will use GRIPP2 as a framework for extracting PPI-related data from the included studies. While GRIPP2 is technically a reporting guideline, it offers clear domains that help us capture how non-scientific actors were involved [13]. The data will be extracted in a summary table and will include title, author, year, design of study, aim, number of co-investigators, levels of involvement, role type, stages, and methods of incorporating non-scientific actors into research. For more information, see Table 1. The data extraction form is based on the GRIPP2 tool.

### 2.6. Risk of Bias (Quality) Assessment

Multiple quality assessment tools are required. To assess the quality of cross-sectional studies we will use the Newcastle–Ottawa scale (NOS). For randomized experimental studies, we will use the Jadad scale (Oxford quality scoring system). For qualitative studies, we will use the Joanna Briggs Institute critical appraisal tool [23]. The Joanna Briggs Institute critical appraisal tool for quasi-experimental studies will be used to assess the quality of quasi-experimental studies, and the Mixed Methods Appraisal Tool (MMAT) scale will be used to assess the quality of mixed-design studies [24].

A pilot study will be carried out to assess the feasibility of the described process for data extraction and quality analysis. This will help evaluate the data collection process using the mentioned tools.

Reviewers will extract data and assess the quality of all included studies, reaching a consensus on their assessment. In case of discrepancies, an independent third reviewer will be consulted. If information is unclear or missing, we will contact the corresponding author of the study.

### 2.7. Strategy for Data Synthesis

Data synthesis with a multilevel approach will be developed in two stages. Firstly, the included studies will be analyzed and synthesized separately according to their design type.

Following established guidelines [25,26], we will synthesize findings through a narrative approach, reporting participant characteristics, interventions, risk of bias, and outcome frequencies. It will be assessed if statistical grouping and meta-analysis are feasible for similar homogeneous outcomes reported in each included study. For qualitative data synthesis, the constant comparative strategy of grounded theory will serve as the framework.

Secondly, findings from different studies will be gathered through thematic analysis of quantitative data [25,26]. For quantitative studies, we will extract numerical results (e.g., participant counts, outcome measures) separately. However, if these studies include textual descriptions (e.g., detailed methods or PPI processes), we will apply a form of thematic analysis to that textual information. In this sense, we treat the ‘qualitative’ portion of quantitative studies in a manner consistent with thematic analysis frameworks [25].

The lead author will have support from two other authors experienced in systematic review methods to ensure a robust data synthesis process that addresses the research question.

The proposed methods will adhere to the PRISMA (Preferred Reporting Items for Systematic Reviews and Meta-Analyses) statement checklist [20]. Specific elements from each of these instruments will be integrated into a checklist.

Status and Timeline of the Study

The overall timeline will follow defined deadlines for each stage. Any amendments or progress updates will be documented and visible in our PROSPERO registration (CRD42024444374). Regular meetings are being held to monitor progress and ensure that the objectives are met.

At present, the study has completed the protocol design, including the establishment of objectives and review questions, the development of systematic search strategies with relevant terms and databases, and the comprehensive literature search, ensuring broad and rigorous coverage of relevant studies. The next phases include the ongoing study selection and assessment through independent review of titles, abstracts, and full texts by the research team, followed by data extraction and synthesis scheduled between January and February 2025, the drafting of results, discussion, and conclusions from March to May 2025, and the dissemination of findings planned for June and July 2025.

## 3. Expected Results

This protocol will provide up-to-date insights into the methodologies and practices of involving non-scientific actors, specifically adults over 65 with malnutrition, at risk of malnutrition or swallowing disorders, as co-researchers in studies. Additionally, it will develop practical guidance for researchers on how to effectively include older adults as non-scientific actors in future studies. While offering valuable insights into participatory research methods, the study acknowledges the potential limitations in generalizing findings due to the focus on malnutrition in this specific population.

Limitations of the study design

The study design has certain limitations that need to be considered. First, the inclusion of studies with diverse methodologies (quantitative, qualitative and mixed) may generate heterogeneity in the data, making it difficult to synthesize and compare results.

Additionally, the target population of the study is restricted to people over 65 years of age with malnutrition or at risk of malnutrition. Although this specificity responds to a critical public health need, the results may not be applicable to other populations or contexts.

Dissemination plans

By publishing this protocol, we ensure methodological transparency and enable critical appraisal prior to conducting the study. The final results will be submitted to peer-reviewed journals of high impact in public health, nutrition, and participatory methods, and presented at specialized conferences. In addition to publishing in peer-reviewed journals and presenting at conferences, we will prepare accessible summaries for older adults and caregivers (e.g., brochures, posters in community centers, brief talks at local senior groups), ensuring that the key findings are shared in formats suitable for the target population. The use of digital platforms and social media will allow for a wider and more effective dissemination of the results.

Amendments and termination of the study

Any amendments to the study protocol will be rigorously documented and will conform to international standards of transparency and reproducibility in research. These amendments will be registered and updated in the Prospective International Register of Systematic Reviews (PROSPERO), ensuring traceability and public access.

In this research, we will implement strategies to assess potential meta-biases, ensuring the validity and robustness of the results. The specific approaches to bias assessment are described below.

-Selective reporting bias: To assess the possible presence of selective reporting within the included studies, a comparison will be made between the registered protocols of the studies and their final publications, if protocols are available. In addition, we will check for inconsistencies or under-reporting of certain outcomes that could indicate a biased selection of data for publication.-Bias in study selection: To assess the possible bias in study selection, we will review whether inclusion and exclusion decisions have been consistent and transparent. To this end, we will maintain a detailed record of the selection process using the COVIDENCE platform, which supports transparent and efficient study screening.

In the event that methodological, operational, or logistical challenges are identified that jeopardize the feasibility of the study, a thorough analysis will be conducted by the research team and stakeholders. If no viable solutions are found, termination of the study will be considered as a last resort. This process will include a detailed documentation of the reasons, together with a final report on the implications for future research.

## Figures and Tables

**Figure 1 healthcare-13-00839-f001:**
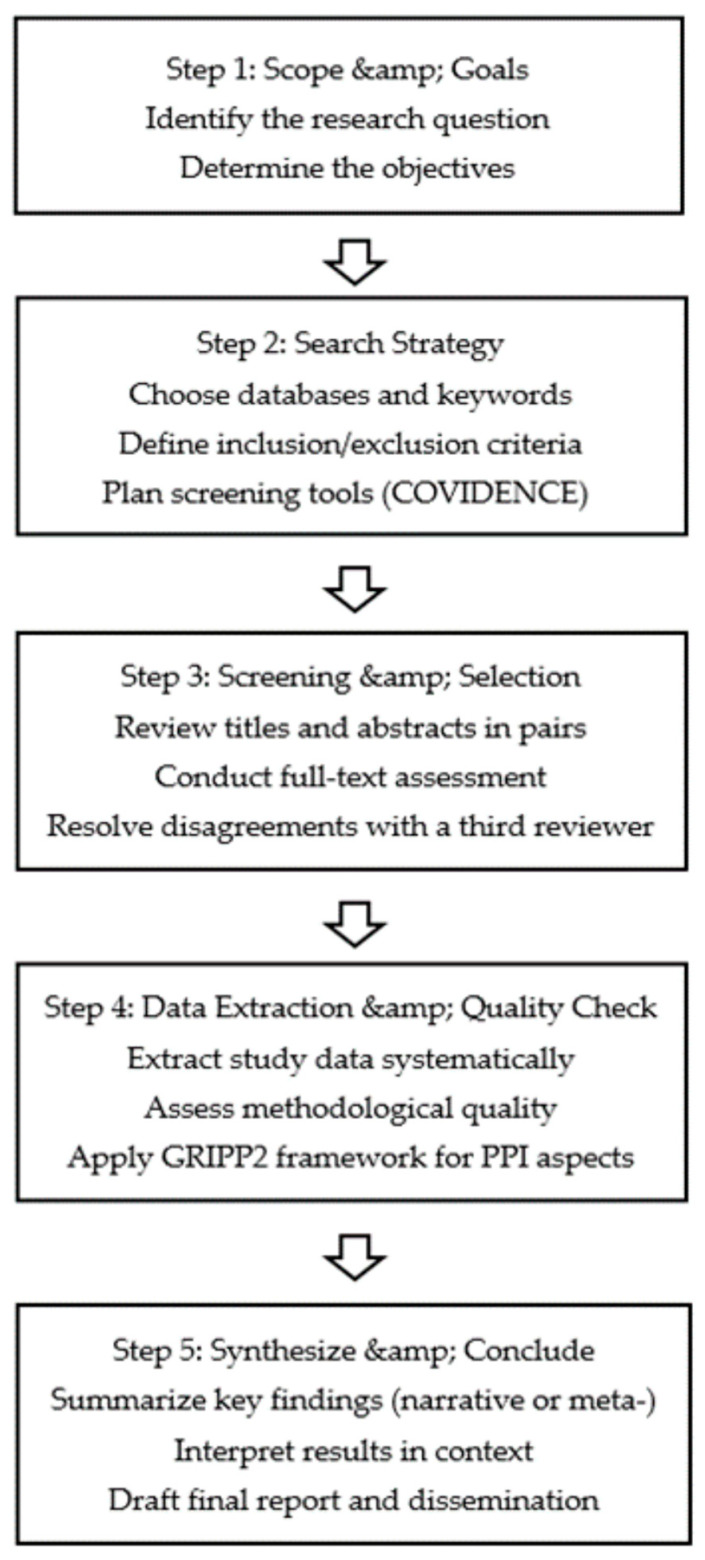
Systematic review process.

**Table 1 healthcare-13-00839-t001:** Data extraction form.

Study, Year	Design	Aim	Number of Co-Investigators ^1^	Levels of Involvement ^2^	Role Type ^3^	Stages ^4^	Methods ^5^	Co-Investigator Profile
Author name/s and publication year	Study design type (qualitative, quantitative, mixed methods, etc.)	Brief description of the study’s primary objective	Total number of individuals involved as co-investigators	Level of involvement according to footnote criteria	Specific role assigned to co-investigators	Research stages where co-investigators participated	Methodological approach used for collaboration	Demographic characteristics and background of co-investigators

^1^ The number of co-researchers included, considering all profiles (academic or community leader, community members, representatives, etc.); ^2^ information (to obtain broad information, opinions, experience, concerning a one-time or specific task question, or topic (i.e., for identification or validation of a topic via a survey)); and consultation (to obtain feedback and advice on a defined research question or research activity (i.e., revise study documents, content relevance, ratings). Patients or the public take an active role in the research project. Other points are collaboration (to work directly with patients throughout or at different moments of the research process to ensure that their expectations and concerns are understood and addressed); partnership (to establish an equal and active co-leadership between the patient and the researcher where decisions about the research process are shared (i.e., members of steering committee or study board)); ^3^ advisor or expert (this role involves patients offering counsel and direction drawn from their individual and collective experiences, representing diverse viewpoints; for example, patients may participate in associations or organizational boards, possessing significant expertise across various facets of the disease care as patient representatives or advocates); personal engagement (personal engagement occurs when individuals, including members of the public not directly affected by the disease, offer their perspectives and feedback rooted in their firsthand experiences); coresearcher (in this capacity, patients are regarded as equal partners possessing indispensable knowledge required for making substantive contributions to the research endeavor); ^4^ stages of the research process in which involvement takes place (identify needs and/or prioritize research topics, study design, development/revision of study documents, methods development, recruitment, data collection, data analysis/results validation, publication (co-author)); ^5^ collaborative design of intervention; facilitation and feedback; learning health collaborative; peer-led intervention is implemented, involving members of the community; and co-led the dissemination, among others [22]. Only items related to PPI methodology are selected for this table.

## Data Availability

The protocol is published in PROSPERO with the registration number CRD42024444374. Any update or modification will be made on this platform. The results will be published once the results have been obtained and analyzed. You may also consult the corresponding author of the article.

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
