# Peer review of "Patient and Public Involvement in Malnutrition Disorders Health Research: A Methodological Systematic Review Protocol"

_healthcare, 2025, doi:10.3390/healthcare13070839_

Round 1
Reviewer 1 Report
Comments and Suggestions for Authors
I consider this protocol “Patient and public participation in malnutrition disorders in health research: a systematic methodological review protocol” to be relevant because malnutrition is a serious problem that worsens morbidity and mortality, particularly affecting people aged 65 and over. With life expectancy increasing, it is urgent to know how to care for and intervene with the elderly population, to improve their health and quality of life. I consider the objective of this protocol to be relevant: to analyze the methodology for involving patients aged 65 and over with malnutrition or at risk of malnutrition as non-scientific actors or co-investigators in research, as they can provide valuable non-technical information on how to support the nutrition of frail people. However, I consider it a challenge to achieve responses to the specific objectives due to the difficulty in measuring the contributions of this involvement, which translates into a significant lack of research dedicated to identifying the best methods for incorporating non-scientific actors. I would like to make some comments that I believe may contribute to the discussion of this protocol:
I completely agree with the limitations of the protocol that have been highlighted, and I recognise the difficulty in generalising results;
I would like to highlight the fact that this protocol seeks to hear what the elderly have to say;
At no point does this protocol describe how malnutrition and the risk of malnutrition are assessed, which in my opinion is an important aspect.
Are you considering including a nutritionist in this team?
Author Response
Comment 1: At no point does this protocol describe how malnutrition and the risk of malnutrition are assessed, which in my opinion is an important aspect.
Response 1: We appreciate this suggestion. In the introduction section, we have added a paragraph describing how malnutrition and the risk of malnutrition are assessed
Old Paragraph
“Conversely, malnutrition can contribute to premature mortality through severe outcomes such as frailty, delirium, reduced immune function, muscle wasting, and cognitive impairment. These factors hinder recovery during illness and diminish patients' quality of life.”
Revised Paragraph, lines 84-90
“Conversely, malnutrition can contribute to premature mortality through severe outcomes such as frailty, delirium, reduced immune function, muscle wasting, and cognitive impairment. These factors hinder recovery during illness and diminish pa-tients' quality of life. Therefore, it is essential to properly assess eating and swallowing disorders using vali-dated clinical and nutritional tools that enable early detection and the implementation of individualized interventions (an-thropometric measurements, biochemical param-eters, MUST, GLIM, EAT-10, among others)”
Comment 2: Are you considering including a nutritionist in this team?
Response 2: A new column has been added to the results collection table: Co-investigator profile. Thus, we will be able to see what types of professionals are used in this type of teams.
Old Table 1
Columns: Study, year; Design; Aim; Number of co-investigators; Levels of involvement; Role type; Stages; Methods.
Revised Table 1, line 404
We now include a column: “Co-investigator Profile” to note whether participants are family caregivers, older adult volunteers, organizational representatives, etc.
Reviewer 2 Report
Comments and Suggestions for Authors
Thanks to the authors for the protocol, which aims to analyse the methodology for involving patients aged 65 and older with malnutrition or at risk of malnutrition as non-scientific actors or co-investigators in research.
There are some comments and suggestions for the protocol:
- The protocol is missing two sections – experimental design and detailed procedure. I would recommend that the authors structure the protocol according to the requirements.
- The text (Lines 65-69) is identical to the text reflected from lines 81 to 84.
- I would recommend that authors present the research design graphically, because otherwise it looks complicated and difficult to understand. This would allow a better understanding of the research process.
- The PICO model consists of four questions: population, intervention, comparison and outcome. Question 3 of the PICO model (comparison) is missing from the protocol (Lines 182-187).
- I would recommend focusing on the last ten years of publication (2014-2024). Authors wrote: without a start date limit (Line 234). As research has evolved significantly, studies older than ten years could change the results of the review.
- There is an error in line 240, the correct is PICO model, not PIC.
- Inaccuracies in the information provided. In section 2.2.2. Population (Lines 210-211), it is stated that studies addressing nutrition (malnutrition, overweight, underweight, etc.) and dysphagia will be included. In line 241, when providing information on the search strategy, only malnutrition, risk of malnutrition, and deglutition disorders are mentioned. Deglutition disorders are broader than dysphagia. So, there is uncertainty.
The submitted manuscript is a protocol that provides information on how the study will be conducted. With the above-mentioned corrections, the protocol provides all the necessary information. From a scientific point of view (after the corrections), it is correct.
My question to the editor is whether this protocol is so valuable that it should be published. The study that will be conducted according to this protocol will be valuable, I have no doubt about that. The authors also state that the results will be published in peer-reviewed scientific journals of high relevance in public health, nutrition, and participatory methods (Lines 365-366).
Author Response
Comment 1: The protocol is missing two sections (experimental design and detailed procedure). It is recommended to structure the protocol according to these requirements.
Response 1:
We appreciate this suggestion. We have included two new sections to fulfill the reviewer’s request and improve clarity:
Experimental Design (or Methodological Design): Describes the rationale and sequence of steps in the review.
Detailed Procedure: Outlines precisely how each phase of the protocol will be conducted (e.g., searching, selecting, extracting, analyzing, and disseminating).
Old Paragraph
“In Section 2 (Materials and Methods), we describe the eligibility criteria, search strategy, study selection, and data analysis. The review follows PRISMA-P guidelines to ensure transparency and reproducibility of the process.”
Revised Paragraph, lines 247 and 305
2.1. Experimental Design
This protocol follows a systematic review design focusing on methodologies for patient and public involvement (PPI). We will first establish the specific objectives and the research question. Next, we will develop the search strategy for the selected databases (PubMed, EMBASE, CINAHL), choose studies based on our inclusion and exclusion criteria, conduct data extraction and quality assessment, and finally synthesize the findings.
2.2. Detailed Procedure
Literature search: We will look through the chosen databases without language restrictions but will apply publication date limits (see Section 2.3.5. Search strategy).
Study selection: Pairs of reviewers will screen titles and abstracts using COVIDENCE.
Data extraction and quality assessment: We will apply recognized tools such as those from JBI and the Newcastle-Ottawa scale, among others.
Synthesis: We will group the included studies (quantitative, qualitative, mixed methods) and employ both a narrative approach and thematic analysis.
Results dissemination: We aim to publish in indexed journals and present at conferences to share outcomes widely.”*
Comment 2: The text in lines 65-69 is repeated in lines 81-84.
Response 2:
We have removed the duplicated content to maintain clarity and avoid redundancy.
Old Paragraph (repeated in lines 67-71 and 91-94)
“Current estimates suggest that about a quarter of older adults are undernourished or at risk of undernutrition, and this number is expected to increase with the rapid growth of the aging population. The increase in the risk of malnutrition may be due to physiological changes associated with aging and specific nutritional needs, among other factors.”
Revised Paragraph, lines 91-94
“Current estimates suggest that about a quarter of older adults are undernourished or at risk of undernutrition, and this number is expected to increase with the rapid growth of the aging population. The increase in the risk of malnutrition may be due to physiological changes associated with aging and specific nutritional needs, among other factors. Tailored counseling, person-centered care, and active implementation strategies are needed to address this problem (1).”
Comment 3: The research design looks complicated; a diagram or figure would improve understanding.
Response 3:
Thank you for this helpful suggestion. We have added a flowchart or roadmap figure in the relevant section to visually represent each step of the systematic review and clarify the involvement of non-scientific actors.
Old Paragraph
“This protocol follows the Preferred Reporting Items for Systematic Reviews and Meta-Analyses for Protocols (PRISMA-P) guidelines.”
Revised Paragraph, lines 254-302
“This protocol follows the Preferred Reporting Items for Systematic Reviews and Meta-Analyses for Protocols (PRISMA-P) guidelines, as illustrated in Figure 1. This figure depicts each stage of the review: (1) defining the research question and objectives, (2) developing the search strategy, (3) screening and selecting studies, (4) data extraction and quality assessment, and (5) synthesizing and concluding. Figure 1 thus provides a clear roadmap for our systematic review process.”
“Figure 1. Systematic review process.”
Comment 4: The PICO model typically has four elements: Population, Intervention, Comparison, and Outcome. Question 3 (Comparison) is missing.
Response 4:
We acknowledge that in our protocol, we do not have a typical “control group” or direct comparison. However, we have clarified this point explicitly, stating why we are not applying the standard “C” in our approach.
Old Paragraph, lines 226-231
“The research question for this review was developed using the PICO format.…”
Revised Paragraph, lines 233-240
“The research question is framed using the PICO model:
P (Population): patients ≥65 of age with impaired nutritional status or deglutition disorders;
I (Intervention): methodologies for including non-scientific actors (PPI) in research;
C (Comparison): not applicable in the traditional sense, as we are not evaluating a control group but rather examining different PPI methods;
O (Outcome): assessment of these methods (using GRIPP2) and the degree of participant involvement.”
Comment 5: It is recommended to concentrate on the last ten years, as research has evolved significantly. Studies older than a decade might affect the review’s outcomes.
Response 5:
Initially, we wanted to include all relevant studies, even those predating 2014, to capture any pioneering approaches. However, we agree that the main evolutions in PPI methodology have taken place in the past decade. Therefore, we will limit our search to articles published from 2014 to 2024.
Old Paragraph, lines 359-360
“The main search will be conducted in each database without a start date limit and will include studies published up to August 2024.”
Revised Paragraph, lines 359-362
“The main search will be conducted in each database, restricting the publication dates to the last ten years (2014-2024) to ensure that we capture the most current and relevant PPI methodologies. We will include articles published up to August 2024.”
Comment 6: There is a typographical error in line 240, which says “PIC” instead of “PICO.”
Response 6:
We have corrected this typo throughout the document, ensuring consistent use of “PICO.”
Old Paragraph
“Search strategy in PIC format (21):”
Revised Paragraph, line 367
“Search strategy in PICO format (21):”
Comment 7: In Section 2.2.2 (Population) (Lines 210-211), it says that studies addressing malnutrition, overweight, underweight, and dysphagia will be included. But in the search strategy (Line 241), only “malnutrition,” “risk of malnutrition,” and “deglutition disorders” are mentioned, and deglutition disorders are broader than dysphagia.
Response 7:
We have aligned these descriptions so that both the “Population” subsection and the search strategy are coherent. Malnutrition indeed encompasses various forms (undernutrition, overweight, etc.), and “deglutition disorders” includes dysphagia.
Old Paragraph
“(Section 2.2.2. Population): It will include studies whose population is over 65 years of age and where nutri-tion (malnutrition, overweight, underweight, etc.) or dysphagia is discussed, and that have incorporated non-scientists into the research.
(Search Strategy): ‘malnutrition,’ ‘risk of malnutrition,’ ‘deglutition disorders’…”
Revised Paragraph, lines 331-334
“(Section 2.2.2. Population): We will include studies involving individuals aged ≥65 years with malnutrition or at risk of malnutrition (covering undernutrition, overweight, or other related nutritional imbalances) and/or deglutition disorders (including dysphagia).
(Search Strategy, Appendix A.1): Our search terms will be ‘malnutrition,’ ‘risk of malnutrition,’ ‘nutrition disorders,’ and ‘deglutition disorders’ (encompassing dysphagia) to ensure comprehensive coverage.”
Comment 8: The reviewer questions whether the protocol itself is sufficiently valuable to publish, though they acknowledge the ultimate study will be valuable.
Response 8:
We genuinely appreciate this perspective. Publishing the protocol is important for transparency, helps avoid selective reporting bias, and offers an opportunity for feedback before carrying out the full study. Since patient and public involvement (PPI) is ever-evolving, sharing explicit protocols helps other teams design and refine their own research.
Old Paragraph, lines 504-506
“The findings will be published in peer-reviewed scientific journals of high rele-vance in public health, nutrition and participatory methods. They will also be pre-sented at conferences to foster interaction with academic and professional audiences.”
Revised Paragraph, lines 506-510
“By publishing this protocol, we ensure methodological transparency and enable critical appraisal prior to conducting the study. The final results will be submitted to peer-reviewed journals of high impact in public health, nutrition, and participatory methods, and presented at specialized conferences.”
Reviewer 3 Report
Comments and Suggestions for Authors
Congratulations to the authors for recognising this important gap in contemporary knowledge and research practice. This will be a very timely report offering much needed guidance to researchers and co-investigator teams. Thank you for sharing your work. My feedback is meant to help strengthen and enhance your article. The suggestions provided are aimed at refining your ideas and ensuring clarity, so your valuable research can de communicated more effectively.
Abstract:
Lines 44-45: Suggestive of a target population of older adults who are frail and with cognitive deficits. However, the rationale for targeting such a population is not supported in the introduction. See my further comments on this below relating to lines 88-90.
Introduction:
Lines 63-74: Missing references to evidence the statements made, e.g. the current estimates of older adults with undernutrition; the ageing population; relationship between undernutrition and frailty, etc. Reference 1 is a review of the literature and not the original research. Citing original research enhances the credibility of the work, as it shows reliance on primary sources rather than secondary interpretations. Relying on multiple original sources allows for a more comprehensive view of the topic, avoiding potential biases or limitations of a single review document.
Lines 81-84: repeat of information already provided in lines 65-69.
Lines 88-90: It is well established that malnutrition and risk of malnutrition is prevalent among the population of older adults. This sentence 'In this regard, it is important to take into account the perspectives of patients and caregivers, as they can provide valuable information on how 89
to support the nutrition of those who are frail or cognitively impaired.' seems to suggest that the perspectives of caregivers in relation to frailty and cognitive impairment are a focus of nutrition care. I suggest that this is a narrow focus and limits the impact of your study.
Lines 93-97: could the authors better describe who they mean by ‘the community’. Perhaps a description in the context of the proposed research question/older population would help with the flow of writing here. Or, as in line 108-109 state ‘the community under study’.
Line 133: is ‘divided’ a better word than ‘uneven’ to describe the varying level of attention given to engaging with GRIPP?
Lines 134-144: a clarification of roles as understood by the authors is required. This observation aligns with my comments on lines 93-97. As currently written, my impression is that ‘the community’, ‘non-scientific actors’, ‘co-researchers’ and ‘co-investigators’ are being used interchangeably. The following citation might help: Hilger, A., Rose, M. & Keil, A. Beyond practitioner and researcher: 15 roles adopted by actors in transdisciplinary and transformative research processes. Sustain Sci 16, 2049–2068 (2021). https://doi.org/10.1007/s11625-021-01028-4
Line 155: Incomplete sentence ‘However, the reality is that most studies…’
Line 155-158: Problem with syntax of this sentence.
Line 164-165: Delete ‘and’ to read ‘Patient and public involvement (PPI) strategies for building relationships include scheduling regular team meetings and reflection sessions maintaining a flexible and interactive process.’
Line 169: suggest the word ‘overall’ replaces ‘main’ and that ‘study’ replaces ‘project’.
Line 169-172: Long sentence that might be better split for clarity.
Line 169-180: A common objective of a systematic review is to provide a comprehensive and unbiased synthesis of all available evidence on a specific research question. It is unusual that there is no specific mention of this motive in the current study.
Materials and methods:
Line 186: Suggest that ‘investigation’ is defined as ‘research’ to align with proposed research question. Investigation could also mean/be confused with clinical procedures given the target population.
Line 188-189: Problem with syntax of this sentence.
Line 194-195: Suggest PRISMA_P checklist is provided as supplementary file.
Line 204-205: ‘could’ should be ‘can’ to keep the same tense throughout the paragraph.
Line 215: Is this a sub-heading?
Line 263: correct ‘qualities’ to read ‘quality of studies’
Line 268-270: are the authors proposing that GRIPP2 provides a framework for data extraction? If so, it may add to the clarity of the proposed methodology to specifically state this.
Table 1: Number of investigators will be recorded but not their profile according to the description provided at subscript 1. Why is profile not recorded? My thoughts are that this would provide contextual information to explain levels of involvement. Profile would seem to be distinct from role type as described in the footnotes to table 1.
Line 314-315: seems to be repetition of lines 309-310
Line 322-327: have the authors got references to support this approach?
Line 328-329: could the authors explain their approach to thematic analysis of quantitative data please. I am familiar with thematic analysis is typically associated with qualitative data, as it involves identifying and analysing patterns or themes within textual data, such as interview transcripts or open-ended survey response, but not quantitative data? The cited references suggest the authors intention to integrate quantitative and qualitative data in the synthesis of the evidence but this could be better elaborated in the current manuscript.
Line 338: Where can the reader observe progress with the proposed timeline. Suggest signposting to PROSPERO.
Expected results:
Line 351: replace ‘it’ with ‘malnutrition’ to read ‘at risk of malnutrition’
Lines 364-369: given the rationale for this study would the authors consider other/alternative/appropriate formats of dissemination with older persons?
Lines 375-376: Problem with syntax of this sentence.
Lines 385-386: suggest rewording ‘with thanks to the COVIDENCE programme’.
Author Response
Comment 1: “Suggestive of a target population of older adults who are frail and with cognitive deficits. However, the rationale for targeting such a population is not supported in the introduction (see further comments on lines 88-90).”
Response 1:
We appreciate this observation. We have clarified in the Introduction why we emphasize frailty and cognitive impairment, including references to studies that highlight the vulnerability of older adults to malnutrition in these contexts. This ensures our rationale is consistent throughout the Abstract and Introduction.
Old Paragraph, lines 43-45
“To address these problems, it is important to take into account the perspectives of patients and caregivers, as they can provide valuable information on how to support the nutrition of frail or cognitively impaired people.”
Revised Paragraph, lines45-48
“To address these challenges, it is important to consider the perspectives of older adults and their caregivers, especially those with conditions such as frailty or cognitive impairment, as they can provide valuable insights on supporting nutrition in these vulnerable populations.”
Comment 2: “Missing references to evidence the statements made (e.g., current estimates of older adults with undernutrition, the aging population, etc.). Reference 1 is a review, not original research. Citing original research enhances credibility.”
Response 2:
We agree. We have replaced or supplemented secondary references with original research studies to substantiate statements on the prevalence of undernutrition, the aging population, and age-related nutritional needs.
Old Paragraph, lines 65-78
“Malnutrition encompasses both overnutrition and undernutrition. … The increase in the risk of malnutrition may be due to physiological changes associated with aging and specific nutritional needs, among other factors (1).”
Revised Paragraph, lines 79-83
“Malnutrition encompasses both overnutrition and undernutrition. Recent evidence indicates that up to 25% of older adults may be undernourished or at risk of undernutrition (1,2). The rapid growth of the aging population, combined with age-related changes and specific nutritional requirements, further contributes to malnutrition risk (3).”
Comment 3: “Repeat of information already provided in lines 65-69.”
Response 3:
We have removed the repeated information to maintain clarity.
Old Paragraph, lines 67-71
[Essentially duplicated text about prevalence and risk factors for malnutrition.]
Revised Paragraph, lines 91-94
[We now have only one concise paragraph about prevalence, ensuring no duplication.]
Comment 4: “Sentence suggests a narrow focus on frailty and cognitive impairment. This might limit the impact of your study.”
Response 4:
We have broadened our language to clarify that while we acknowledge the importance of frailty and cognitive impairment, our study aims to include perspectives of all older adults at risk of malnutrition, not solely those who are frail or cognitively impaired.
Old Paragraph, lines 98-100
“In this regard, it is important to take into account the perspectives of patients and caregivers, as they can provide valuable information on how to support the nutrition of those who are frail or cognitively impaired.”
Revised Paragraph, lines 100-103
“In this regard, it is crucial to incorporate the perspectives of patients and caregivers across a range of conditions, including but not limited to frailty or cognitive impairment, as they can offer diverse insights into supporting older adults’ nutritional needs.”
Comment 5: “Could the authors better describe who they mean by ‘the community’? A description in the context of the older population would help.”
Response 5:
We have clarified what we mean by “the community,” explaining that it refers to older adults, their informal caregivers, patient associations, and relevant stakeholders who might not have formal scientific roles but are directly impacted by malnutrition.
Old Paragraph, lines 111-113
“Community members may not have research skills, but they can be part of the decision-making process…”
Revised Paragraph, lines 113-118
“In this protocol, ‘the community’ refers to older adults themselves—whether they are living independently or in care facilities—as well as informal caregivers, family members, patient associations, and other stakeholders directly affected by malnutrition who may not hold formal scientific roles. These community actors can still play a critical part in shaping the study design, data interpretation, and dissemination of results.”
Comment 6: “Is ‘divided’ a better word than ‘uneven’?”
Response 6:
Yes, we changed the word “uneven” to “divided” to improve clarity.
Old Sentence
“However, this methodology has received uneven attention across scientific projects, depending on the context.”
Revised Sentence, line 152
“However, this methodology has received divided attention across scientific projects, depending on the context.”
Comment 7: “Clarification of roles is needed. ‘the community,’ ‘non-scientific actors,’ ‘co-researchers,’ and ‘co-investigators’ appear interchangeable. Also consider referencing Hilger et al. (2021).”
Response 7:
We have added a brief definition for each term and included the suggested citation to reinforce how roles can vary in transdisciplinary research.
Old Paragraph
“There remains a significant lack of research dedicated to identifying the best methods for incorporating non-scientific actors. … It is crucial to develop and disseminate strategies that enhance researchers’ preparedness to collaborate with communities.”
Revised Paragraph, lines 153-167
“There remains a significant lack of research describing how best to include non-scientific actors, a term we use here to encompass diverse stakeholders such as family caregivers, patient associations, or community representatives. In some cases, these partners serve as co-researchers, offering consistent input throughout the project; at other times, they become co-investigators, indicating a deeper level of shared decision-making with the academic team. By clarifying these roles, we can develop and disseminate strategies that enhance researchers’ preparedness to collaborate effectively.”
Comment 8: “Incomplete sentence and syntax problems.”
Response 8:
We have restructured these sentences to correct any syntax issues and improve clarity.
Old Sentence
“However, the reality is that most studies…”
Revised Sentence, lines 179-181
“However, in reality, most studies involving older adults and malnutrition provide limited details on how non-scientific actors are recruited or integrated into the research process.”
Comment 9: “Delete ‘and’ to read ‘…reflection sessions maintaining a flexible…’”
Response 9:
We removed the extra “and” to maintain consistent sentence structure.
Old Sentence
“Patient and public involvement (PPI) strategies for building relationships include scheduling regular team meetings and reflection sessions and maintaining a flexible and interactive process.”
Revised Sentence, lines189-192
“Patient and public involvement (PPI) strategies for building relationships include scheduling regular team meetings and reflection sessions, maintaining a flexible and interactive process.”
Comment 10: “Suggest ‘overall’ replaces ‘main’ and ‘study’ replaces ‘project’.”
Response 10:
We made these wording changes to better reflect the nature of the systematic review.
Old Sentence
“The main objective of this project is to analyse the methodology for involving patients aged 65 and older…”
Revised Sentence, line 198
“The overall objective of this study is to analyse the methodology for involving patients aged 65 and older…”
Comment 11: “Long sentence—split for clarity. Also unusual that there is no mention of providing a comprehensive synthesis of available evidence, which is a common objective for systematic reviews.”
Response 11:
We split the sentence and added a direct mention of the aim to synthesize available evidence on involving older adults as co-researchers, but due to another reviewer's comment we have finally changed it to describe the objectives of the protocol, instead of the systematic review.
Old Sentence
“The main objective of this project is to analyse the methodology for involving patients aged 65 and older with malnutrition or at risk of malnutrition as non-scientific actors or co-investigators in research and to make proposal plan and evaluate the incorporation of non-scientific actors.”
Revised Paragraph, 198-202
“The overall objective of this study is to analyse the methodology for involving patients aged 65 and older with, or at risk of, malnutrition as co-investigators in research. We also aim to provide a comprehensive and unbiased synthesis of available evidence, developing a practical guide to inform future studies on how to effectively incorporate non-scientific actors.”
Revised Paragraph, assuming the changes from another comment, lines 212-224
By publishing this protocol, we aim to offer a clear methodological roadmap for researchers interested in systematically involving older adults at risk of malnutrition as co-investigators. While the final outcomes will be presented in subsequent publica-tion, this protocol provides immediate value by:
- Ensuring methodological rigor and transparency, thus reducing the risk of se-lective reporting.
- Allowing other researchers to replicate or adapt our approach to different populations and contexts.
- Establishment of a foundational framework for patient and public involve-ment (PPI) in nutritional and/or swallowing disorders research.
In doing so, we hope to enhance the quality and relevance of future research on malnutrition, bridging the gap between the academic sphere and the real-world per-spectives of older adults and their caregivers.
Comment 12: “Suggest that ‘investigation’ is defined as ‘research’ to align with the research question.”
Response 12:
We replaced “investigation” with “research” to remain consistent.
Old Wording
“Types of intervention/phenomena of interest: The phenomenon of interest in this review will be the methodology used for the inclusion and involvement of non-scientific actors in the different stages of an investigation.”
Revised Wording, lines 338-340
“Types of intervention/phenomena of interest: The phenomenon of interest in this review will be the methodology used to include and involve non-scientific actors in various stages of a research study.”
Comment 13: “Problem with syntax of this sentence.”
Response 13:
We have rephrased it for clarity.
Old Sentence
“Studies that involve non-scientific actors, as well as those that describe the strategies and approaches used to facilitate the incorporation of this group into research will be included.”
Revised Sentence, lines 342-344
“We will include studies that incorporate non-scientific actors or describe the strategies used to enable their involvement in research.”
Comment 14: “Suggest PRISMA-P checklist is provided as a supplementary file.”
Response 14:
We now include the PRISMA-P checklist in the Supplementary Materials to allow readers to see how the protocol aligns with best practices.
Old Paragraph
“This protocol has been developed in accordance with the Preferred Reporting Items for Systematic Reviews and Meta-Analyses (PRISMA-P) checklist.”
Revised Paragraph, lines 260-261
“This protocol follows the Preferred Reporting Items for Systematic Reviews and Meta-Analyses for Protocols (PRISMA-P) guidelines. The complete PRISMA-P checklist is provided as a Supplementary File for transparency.”
Comment 15: “Use ‘can’ instead of ‘could’ for consistent tense.”
Response 15:
We have changed the phrasing to maintain consistency in tense.
Old Sentence
“Qualitative studies or mixed methods in which data according to the aim of the review could be extracted.”
Revised Sentence, line 323
“Qualitative or mixed-method studies in which data can be extracted according to the review’s aim…”
Comment 16: “Is this a sub-heading?”
Response 16:
Yes, we have re-labeled it clearly as a sub-heading or moved it to ensure logical structure.
Old Placement
(A single line that was ambiguous and not clearly formatted.)
Revised Placement, line 345
We now present it as a proper sub-heading, for instance: “2.3.3. Context” (or whichever label) so it’s clearly identified.
Comment 17: “Correct ‘qualities’ to read ‘quality of studies’.”
Response 17:
We fixed this wording.
Old Text
“To extract data and assess the qualities of the studies, the research team members will work in pairs…”
Revised Text, line 390
“To extract data and assess the quality of the studies, the research team members will work in pairs…”
Comment 18: “Are the authors proposing that GRIPP2 provides a framework for data extraction? If so, clarify this.”
Response 18:
Yes, we are using GRIPP2 to guide extraction of information specifically related to patient/public involvement. We now explicitly state this in the text.
Old Text
“GRIPP2 was considered to extract data related to PPI from the included studies, despite being a reporting guideline tool.”
Revised Text, lines 397-399
“We will use GRIPP2 as a framework for extracting PPI-related data from the included studies. While GRIPP2 is technically a reporting guideline, it offers clear domains that help us capture how non-scientific actors were involved.”
Comment 19:
“Number of investigators will be recorded but not their profile. Why not record their profile? It could provide contextual info.”
Response 19:
We agree that collecting profile data would enrich our context. We have added an additional column to capture the profile of the co-investigators (e.g., caregiver, community member, etc.).
Old Table 1
Columns: Study, year; Design; Aim; Number of co-investigators; Levels of involvement; Role type; Stages; Methods.
Revised Table 1, line 404
We now include a column: “Co-investigator Profile” to note whether participants are family caregivers, older adult volunteers, organizational representatives, etc.
Comment 20: “Seems to be repetition of lines 309-310.”
Response 20:
We removed the duplicated text, streamlining this discussion.
Old Paragraph
(Essentially a restatement of the same idea about data synthesis.)
Revised Paragraph, lines 439-440
[We now provide just one concise explanation of the approach to data synthesis, avoiding repetition.]
Comment 21: “Have the authors got references to support this approach?”
Response 21:
We have now included additional references that justify our synthesis methods, showing how other systematic reviews combine different study designs.
Old Paragraph
“Studies will be synthesized descriptively through narratives, reporting on participant characteristics, intervention compositions, risk of bias in outcomes, and outcome frequencies.”
Revised Paragraph, lines 454-456
“Following established guidelines (25, 26), we will synthesize findings through a narrative approach, reporting participant characteristics, interventions, risk of bias, and outcome frequencies.”
Comment 22: “Explain thematic analysis of quantitative data. Typically thematic analysis is used for qualitative data.”
Response 22:
We have clarified that we do not apply thematic analysis directly to numerical quantitative outcomes but rather to textual descriptions of methodology and PPI reports in quantitative studies.
Old Paragraph
“Additionally, exploration will be conducted using other tools and techniques to construct a common rubric for a global thematic synthesis.”
Revised Paragraph, lines 462-467
“For quantitative studies, we will extract numerical results (e.g., participant counts, outcome measures) separately. However, if these studies include textual descriptions (e.g., detailed methods or PPI processes), we will apply a form of thematic analysis to that textual information. In this sense, we treat the ‘qualitative’ portion of quantitative studies in a manner consistent with thematic analysis frameworks (25).”
Comment 23: “Where can the reader observe progress with the proposed timeline? Suggest signposting to PROSPERO.”
Response 23:
We now indicate that the timeline and any updates will be noted in PROSPERO.
Old Paragraph
“The overall timeline will follow the deadlines set for each stage…”
Revised Paragraph, lines 475-477
“The overall timeline will follow defined deadlines for each stage. Any amendments or progress updates will be documented and visible in our PROSPERO registration (CRD42024444374).”
Comment 24:
“Replace ‘it’ with ‘malnutrition’ to read ‘at risk of malnutrition’.”
Response 24:
We have made that small correction to improve clarity.
Old Text
“…specifically adults over 65 with malnutrition or at risk of it, as co-researchers…”
Revised Text, line 490
“…specifically adults over 65 with malnutrition, at risk of malnutrition or swallowing disorders, as co-researchers…”
Comment 25: “Consider other/alternative dissemination formats with older persons.”
Response 25:
We agree and have expanded our dissemination plans to include more accessible formats, such as brochures, community talks, or local radio segments that cater to older individuals.
Old Paragraph
“The findings will be published in peer-reviewed scientific journals of high relevance in public health, nutrition, and participatory methods. They will also be presented at conferences…”
Revised Paragraph, lines 510-513
“In addition to publishing in peer-reviewed journals and presenting at conferences, we will prepare accessible summaries for older adults and caregivers (e.g., brochures, posters in community centers, brief talks at local senior groups), ensuring that key findings are shared in formats suitable for the target population.”
Comment 26: “Problem with syntax of this sentence.”
Response 26:
We revised the sentence for better flow.
Old Sentence
“This research, strategies for the assessment of potential meta-biases will be implemented to ensure the validity and robustness of the results.”
Revised Sentence, lines 522-524
“In this research, we will implement strategies to assess potential meta-biases, ensuring the validity and robustness of the results.”
Comment 27: “Suggest rewording ‘with thanks to the COVIDENCE programme.’”
Response 27:
We have rephrased it more gracefully.
Old Sentence
“A detailed record of the selection process will be maintained thanks to the COVIDENCE programme.”
Revised Sentence, lines 533-535
“We will maintain a detailed record of the selection process using the COVIDENCE platform, which supports transparent and efficient study screening.”
Reviewer 4 Report
Comments and Suggestions for Authors
The manuscript has a good structure, but I can't see the manuscript can add some value or insight without results. In my opinion, the research gain is not clear uns should be better described.
Author Response
Comment 1: There is nothing wrong with the structure, the writing or the method of the manuscript. But I can’t see any new information/ knowledge within the manuscript. Right now, it seems like a paper without results, which is not readable (in my opinion). If the method is something new, it should be described as a result and also discussed why this is new and better than other methods.
Response 1:
We appreciate this feedback. As a protocol, this manuscript is designed to outline our systematic approach, methodology, and rationale before we conduct the actual review and synthesize results. Publishing the protocol in advance helps ensure methodological transparency, prevents selective reporting bias, and allows the broader research community to critique and refine our approach.
To address the reviewer’s point about “research gain,” we have expanded the “significance” or “research gain” section. We highlight why documenting our methodological framework is valuable: it lays groundwork for reproducibility, invites critical evaluation of our methods, and clarifies how future researchers can adapt these methods for patient and public involvement (PPI) studies.
Old Paragraph
“The main objective of this project is to analyze the methodology for involving patients aged 65 and older with malnutrition or at risk of malnutrition as co-investigators in research.”
Revised Paragraph, 212-224
“By publishing this protocol, we aim to offer a clear methodological roadmap for researchers interested in systematically involving older adults at risk of malnutrition as co-investigators. While the final outcomes will be presented in subsequent publication, this protocol provides immediate value by:
Ensuring methodological rigor and transparency, thus reducing the risk of selective reporting.
Allowing other researchers to replicate or adapt our approach to different populations and contexts.
Establishment of a foundational framework for patient and public involvement (PPI) in nutritional and/or swallowing disorders research.
In doing so, we hope to enhance the quality and relevance of future research on malnutrition, bridging the gap between the academic sphere and the real-world perspectives of older adults and their caregivers.”
Round 2
Reviewer 3 Report
Comments and Suggestions for Authors
Thank you for returning a detailed response to review. I agree with the changes made. A typo noted in box 1 of Figure 1 but this will likely be picked up in proofs. Best of luck with this publication.
Author Response
Comments 1: Thank you for returning a detailed response to review. I agree with the changes made. A typo noted in box 1 of Figure 1 but this will likely be picked up in proofs. Best of luck with this publication.
Response 1: We sincerely value your comments and your support of our publication. We have reviewed Figure 1 and corrected the typographical error in Box 1. We have also eliminated the typographical errors in the rest of the boxes, since they had the same. We truly appreciate your attention to detail and the time you have dedicated to reviewing our work.
Former text
“Step 1: Scope & Goals” ” Step 3: Screening & Selection” “Step 4: Data Extraction & Quality Check” “Step 5: Synthesize & Conclude”
Revised text, lines 266, 280, 288, 296
“Step 1: Scope & Goals” ” Step 3: Screening & Selection” “Step 4: Data Extraction & Quality Check” “Step 5: Synthesize & Conclude”
Reviewer 4 Report
Comments and Suggestions for Authors
The authors made the manuscript goals and possible readers interest more clear. Accordingly there is nothing to improve.
Author Response
Comments 1: The authors made the manuscript goals and possible readers interest more clear. Accordingly there is nothing to improve.
Response 2: We sincerely appreciate your positive feedback regarding the manuscript. We are pleased to hear that the goals and potential readers’ interest are now clearer. Thank you for your time and valuable input.